# Integrated Bioinformatics Analysis of the Hub Genes Involved in Irinotecan Resistance in Colorectal Cancer

**DOI:** 10.3390/biomedicines10071720

**Published:** 2022-07-16

**Authors:** Jakub Kryczka, Joanna Boncela

**Affiliations:** Institute of Medical Biology, Polish Academy of Sciences, 93-232 Lodz, Poland; jkryczka@cbm.pan.pl

**Keywords:** CRC, irinotecan resistance, multidrug resistance, chemotherapy, integrated bioinformatics analysis

## Abstract

Different drug combinations including irinotecan remain some of the most important therapeutic modalities in treating colorectal cancer (CRC). However, chemotherapy often leads to the acquisition of cancer drug resistance. To bridge the gap between in vitro and in vivo models, we compared the mRNA expression profiles of CRC cell lines (HT29, HTC116, and LoVo and their respective irinotecan-resistant variants) with patient samples to select new candidate genes for the validation of irinotecan resistance. Data were downloaded from the Gene Expression Omnibus (GEO) (GSE42387, GSE62080, and GSE18105) and the Human Protein Atlas databases and were subjected to an integrated bioinformatics analysis. The protein–protein interaction (PPI) network of differently expressed genes (DEGs) between FOLFIRI-resistant and -sensitive CRC patients delivered several potential irinotecan resistance markers: NDUFA2, SDHD, LSM5, DCAF4, COX10 RBM8A, TIMP1, QKI, TGOLN2, and PTGS2. The chosen DEGs were used to validate irinotecan-resistant cell line models, proving their substantial phylogenetic heterogeneity. These results indicated that in vitro models are highly limited and favor different mechanisms than in vivo, patient-derived ones. Thus, cell lines can be perfectly utilized to analyze specific mechanisms on their molecular levels but cannot mirror the complicated drug resistance network observed in patients.

## 1. Introduction

Colorectal cancer (CRC) is the third most common cancer worldwide and the second leading cause of cancer death [1]. The main therapeutic approaches to treat CRC are surgery, chemotherapy, irradiation, and targeted therapy [2]. Combined with 5-fluorouracil (5-FU), irinotecan is considered the standard therapeutic agent of the first-line treatment, the FOLFIRII regimen. Combination chemotherapy regimens have been shown to achieve greater efficacy than single-agent treatments in CRC [3]. After more than two decades of clinical usage, irinotecan has become a versatile chemotherapeutic agent that combines well not only with 5-FU but also with monoclonal antibodies, such as cetuximab and bevacizumab. Experimental and clinical studies have also indicated that irinotecan could be used with kinase inhibitors, such as fruquintinib, apatinib, dasatinib, regorafenib, and sunitinib, or with cell-cycle checkpoint inhibitors [4,5]. Although irinotecan still represents the backbone of CRC chemotherapy, drug resistance is a significant obstacle that severely limits the efficacy of irinotecan and often causes cancer relapse and metastasis [6]. 

The mechanism of irinotecan resistance is still inconclusive and requires further investigation [7]. Since the intracellular concentration of irinotecan or its active metabolites can be reduced by increased efflux or increased metabolism, the cellular processing and irinotecan transport out of cancer cells can be exploited to identify biomarkers of therapeutic failures.

Irinotecan is a prodrug that requires bioactivation to form the active metabolite SN-38. SN-38 inhibits topoisomerase-I (Topo 1) activity and affects tumor cell proliferation. The bioactivation of irinotecan occurs rapidly after administration in the plasma and liver. Irinotecan is subjected to enzymatic cleavage performed by two isoforms of carboxylesterases (CES1 and CES2) and butyrylcholinesterase (hBChE) that removes bulky side chains, forming SN-38, which presents 100- to 1000-fold greater activity than irinotecan itself. CES1 and CES2 are predominantly active in the liver, colon, kidneys, and blood cells. In contrast, the activity of hBChE is mainly localized in blood plasma [7,8]. SN-38 may be further conversed by uridine diphosphate-glucuronosyl transferases (UGT) from the UGT1 family, such as UGT1A1 and UGT1A9 in the liver or extrahepatic UGT1A1, UGT1A7, and UGT1A10 in bile, into inactive SN-38 glucuronide (SN-38G), which in turn may be further processed by the intestinal bacterial β-glucuronidase back into active SN-38 [9,10]. The human UGT1 gene is located on chromosome 2q37, producing nine functional enzymes (UGT1A1, UGT1A3, UGT1A4, UGT1A5, UGT1A6, UGT1A7, UGT1A8, UGT1A9, and UGT1A10) by exon-sharing, possessing similar substrates but differently expressed depending on tissue type [11]. Irinotecan, like most the xenobiotics, is oxidized by CYP3A4 and CYP3A5—members of the cytochrome P450 superfamily—into inactive aminopentane carboxylic acids, APC (7-ethyl-10[4-N-(5-aminopentanoicacid)-1-piperidino] carbonyloxycamptothecin), and NPC (7-ethyl-10[4-amino-1-piperidino]carbonyloxycamptothecin). NPC is a weak inhibitor of cell growth, but contrary to APC, NPC can be further reprocessed into active SN-38 by CES1 and CES2 [12,13]. In addition to metabolism, another critical aspect of the irinotecan body’s processing is its transport out of cells executed by transporters belonging to the ATP-binding cassette (ABC) protein superfamily [14]. Irinotecan and SN-38 are transported into blood or bile from hepatocytes by ABCB1, ABCC1, ABCC2, and ABCG2, whereas the OATP transporter (SLCO1B1) enables its influx from blood [15,16,17]. The active efflux of irinotecan and its metabolites by ABC proteins is also responsible for acquired multidrug resistance (MDR). ABC proteins, such as ABCB1, ABCB5, ABCC1, ABCC2, ABCC4, ABCC5, and ABCG2, have been identified as being involved in irinotecan and SN-38 transport out of the cytosol to reduce its intracellular concentration and efficacy in cancer cells [18,19]. Although enzymes and transporters involved in the irinotecan body’s metabolism and disposition are known, their contributions to irinotecan resistance are still poorly understood.

In vitro studies based on the molecular analysis of drug-sensitive cell lines vs. drug-resistant cell lines have been commonly used to understand the irinotecan resistance mechanism in CRC [20,21]. Cell lines are a less expensive and less complicated model than primary cells that provide an unlimited supply of reproducible material and bypass ethical concerns regarding the usage of patient-derived samples [22,23,24]. Drug-resistant variants of specific cell lines are obtained through continuous exposure to increasing concentrations of irinotecan itself [21] or its active metabolite, SN-38 [20], in growth media for several months. The complicated natures of irinotecan activation, deactivation, and re-activation, followed by its transport and bioavailability, create substantial discrepancies between the knowledge about cancer biology generated in cell line experiments and the translation of this knowledge into clinically useful information [25]. 

This study uses microarray gene expression profiles from an open database to identify potential biomarkers for irinotecan resistance in CRC. We analyze the expression profiles of the primary genes involved in the irinotecan metabolism and transport out of the cells. We aim to investigate mRNA profile differences between several CRC cell lines commonly used in in vitro studies on irinotecan resistance, such as HT29, LoVo, and HTC116, as well as their irinotecan (SN-38)-resistant variants to highlight the heterogeneity of the acquired irinotecan resistance mechanism. Our main objective, and the novelty approach, is to use changes in the molecular patterns of patient-derived samples to validate the usefulness of popular in vitro CRC models. Thus, we analyze the mRNA profiles of CRC samples obtained from patients with metastatic colon cancer who are sensitive or resistant to the first-line treatment of FOLFIRI. Next, we select the top 10 networking genes and proteins (five upregulated and five downregulated) using a protein-protein interaction network of differently expressed genes (DEGs) that may be considered potential biomarkers of irinotecan resistance in vivo. Finally, we validate irinotecan resistance-related DEG expression in obtained irinotecan (SN-38)-resistant CRC cell lines, providing their in-depth molecular analyses of resistance-related mechanisms.

## 2. Materials and Methods

### 2.1. Microarray Data Processing and Analysis

Gene Expression Omnibus (GEO) database (http://www.ncbi.nlm.nih.gov/geo/ (accessed on 18 May 2022)) gene expression profiles were downloaded with accession numbers GSE42387, GSE62080, and GSE18105. The cell lines and their irinotecan (SN-38)-resistant variants were obtained as described [26] Briefly, HCT116, HT29, and LoVo cells were exposed in vitro to gradually increasing SN-38 concentrations for about nine months, generating sub-cell lines with acquired resistance. Gene expression profiles of the parental and resistant cell lines were obtained after 2–3 weeks of culturing in a drug-free medium (each in triplicate) using a GPL16297 Agilent-014850 Whole Human Genome Microarray 4x44K G4112F platform (Agilent Systematic Name, collapsed probe, version). The GSE62080 database comprised 21 patients with advanced colorectal cancer treated using the FOLFIRI scheme and classified according to the WHO criteria in responder (sensitive (S)) and nonresponder (resistant (R)) groups. Detailed information on patient clinical-pathological characteristics was presented in the original paper (https://www.ncbi.nlm.nih.gov/pmc/articles/PMC2257989/table/T2/?report=objectonly (accessed on 18 May 2022)) [27]. Gene expression profiles were obtained using Human Genome GeneChip arrays U133. The GSE18105 database included mRNA profiles of normal adjacent tissue and primary CRC tumors from different stages. One hundred and eleven microarray datasets (77 for LCM samples and 17 pairs for homogenized samples from tumors and adjacent tissues) were normalized through a robust multi-array average (RMA) method using R 2.6.2 statistical software with the BioConductor package. Next, the gene expression levels were log2-transformed. We excluded samples named “metastatic recurrence” obtained from homogenized normal tissue for our analysis. All the data were processed using the GEO2R online analytical tool, which uses the R language (https://www.ncbi.nlm.nih.gov/geo/geo2r/ (accessed on 18 May 2022)) [28]. Differently expressed genes (DEGs) were further calculated and visualized with JASP 0.14.1.0 software (https://jasp-stats.org/ (accessed on 18 May 2022)) using a normality test (Shapiro–Wilk’s), followed by a Mann–Whitney U test (for not normally distributed data) or *T*-test (for normally distributed data). Additionally, in the case of a small n size, we used JASP 0.14.1.0 software (https://jasp-stats.org/ (accessed on 18 May 2022)) to perform a Bayesian Mann–Whitney U test based on a data augmentation algorithm with five chains of 1000 iterations to verify our data further.

### 2.2. Hierarchical Clustering Analysis

After extracting the expression values from the gene expression profiles, a bidirectional hierarchical clustering heatmap was constructed using the Orange open-source machine learning and data visualization platform (https://orangedatamining.com/ (accessed on 18 May 2022)).

### 2.3. Construction of PPI Network

A protein–protein interaction (PPI) network of differently expressed genes (DEGs) was created using STRING version 11.0 online software (https://string-db.org/ (accessed on 18 May 2022)) and an open-source software platform for visualizing complex networks called Cytoscape (https://cytoscape.org/ (accessed on 18 May 2022)). A KEGG (Kyoto Encyclopedia of Genes and Genomes) pathway analysis was performed using the DAVID online tool (https://david.ncifcrf.gov/ (accessed on 18 May 2022)) and the KEGG PATHWAY database (https://www.kegg.jp/kegg/pathway.html (accessed on 18 May 2022)).

All URL address and obtained data were validated and accessed on 18 May 2022.

## 3. Results

We analyzed the mRNA profiles of three CRC cell lines (HT29, LoVo, and HTC116) and their respective irinotecan (SN-38)-resistant variants using the GSE42387 database. Gene expression profiles of the parental and resistant cell lines were obtained from cells cultured for 2–3 weeks in a drug-free medium, each in triplicate (https://www.ncbi.nlm.nih.gov/geo/query/acc.cgi?acc=GSE42387 (accessed on 18 May 2022)). In our analysis, we first focused on ABC proteins, as their expression levels are considered to have a high impact on the overall SN-38 intracellular concentration and are mainly analyzed in CRC cells lines to establish the mechanism of acquired irinotecan resistance [27,29], as shown in Figure 1A. Then, we analyzed the expression levels of enzymes involved in irinotecan and SN-38 metabolism, as shown in Figure 1B. Since our previous study showed that the expression of ABC transporters in CRC was correlated to cell phenotype and shifts during ongoing EMT [18] and considering that many other reports have shown that irinotecan resistance is often related to advanced EMT, we additionally compared the expression levels of significant EMT markers, such as E-cadherin (CDH1), N-cadherin (CDH2), and vimentin (VIM) [30], as shown in Figure 1C.

Our analysis proved the high heterogeneity among all the tested cell lines. All the tested SN-38-resistant variants of CRC cell lines presented elevated expressions of several ABC proteins; however, the profiles differed among all the cell lines, as seen in Figure 1A. The expressions of main SN-38 and irinotecan transporters, i.e., ABCB1, ABCC1, ABCC2, and ABCG2, were significantly upregulated in most SN-38-resistant variants, except for HCT116 cells. HTC116 SN-38 manifested ABCC4 upregulation. Our previous study showed that, in CRC, the elevated expression of ABCC4 was correlated with EMT and the mesenchymal phenotype [18]. However, HT29-SN-38- and LoVo-SN-38-resistant cells, which presented an evident mesenchymal phenotype (downregulation of CDH1 accompanied by upregulation of vimentin and CDH2), exhibited higher mRNA expressions of ABCG2. Interestingly, HT29-SN-38-resistant cells showed the upregulation of ABCC5, which strongly resembled the structure and substrate specificity of ABCC4 [19]. Similar to our findings, the irinotecan (not SN-38)-resistant CRC cell line S1-IR20 presented an upregulated level of protein expression for ABCG2, but not ABCB1 or ABCC1, compared to those of parental S1 cells [21].

Regarding the expression profiles of the main enzymes involved in irinotecan processing, we noticed that HT29-SN-38-resistant variants demonstrated increased CES1 and decreased CES2 expressions. Surprisingly, HCT116-SN-38-resistant variant cells presented an upregulated mRNA level of CES2 and a downregulated mRNA level of CYP3A5 that metabolized SN-38 into NPC. The mRNA levels of UGT1 enzymes, which inactivate SN-38 into SN-38G, also presented ambiguous results. UGT1A6 was upregulated in HT29- and LoVo-SN-38-resistant variants; UGT1A8 was upregulated in HT29 SN-38 but downregulated in LoVo SN-38 cells.

To summarize changes between different cell lines and their SN-38-resistant variants, we used the mRNA levels of the above-analyzed genes as an algorithm to perform a clustering analysis via Orange, an open-source machine learning and data visualization platform, as shown in Figure 2. Our results indicated that all the SN-38-resistant variants presented significantly different profiles than their respective parental cell lines; however, the HT29 cell line and its SN-38-resistant variant differed significantly from the other tested cell lines and clustered separately. Interestingly, using the changes in levels of the above-mentioned mRNA as an algorithm, the LoVo cell line was more closely related to HCT116 and HTC116 SN-38 than to the LoVo SN-38 variant; thus, we can assume that acquired resistance for SN-38 triggered the most significant changes in the mRNA expression profiles of LoVo.

Next, we analyzed samples from 21 patients with advanced colorectal tumors (GSE62080; https://www.ncbi.nlm.nih.gov/geo/query/acc.cgi?acc=GSE62080 (accessed on 18 May 2022)) that, at the end of a first-line treatment FOLFIRI scheme were attributed to responders (sensitive (S)) and nonresponders (resistant (R)) groups according to the WHO criteria [27]. The expression profiles of genes related to chemotherapy resistance observed in CRC tissue samples differed substantially from those obtained from the resistant cell line analysis. In the patient samples, no statistically significant changes were observed in the mRNA expression levels of either ABC transporters or irinotecan-processing enzymes between the sensitive and resistant patient groups. Surprisingly, the mRNA levels of vimentin (EMT marker) and TWIST-1 (EMT-related transcription factor) were significantly downregulated in the resistant patient group (Appendix A). To better characterize the patient cohort, we analyzed the expression of leucine-rich repeat-containing G-protein-coupled receptor 5 (LGR5) [31]. The Bayesian Mann–Whitney U test BF10 suggested “anecdotal evidence” of mRNA upregulation in resistant groups (data not shown). LGR5 potentiates canonical Wnt/β-catenin signaling and is considered a marker of advanced CRC state and a highly unfavorable prognostic marker [31].

Next, we analyzed the top 250 differently expressed genes (DEGs) between the resistant and sensitive patient groups using a protein–protein interaction (PPI) network with STRING version 11.0 (https://string-db.org/ (accessed on 18 May 2022)), as shown in Figure 3. The PPI network was enhanced by known proteins that filled the gaps in proper networking, as described in [28]. The upgraded networking contained: for upregulated genes, number of nodes: 198; number of edges: 2141; average node degree: 1.6; average local clustering coefficient: 0.64; expected number of edges: 488; and PPI enrichment *p*-value: <1.0e-16; and for downregulated genes, number of nodes: 203; number of edges: 2989; average node degree: 29.4; avg. local clustering coefficient: 0.656; expected number of edges: and 756; PPI enrichment *p*-value: <1.0e-16. First, we performed a KEGG pathway analysis using the KEGG PATHWAY database (https://www.kegg.jp/kegg/pathway.html (accessed on 18 May 2022)), STRING version 11.0, and the DAVID online tool. This analysis indicated several signaling pathways that reached a false discovery rate (FDR) < 0.05 and a *p* < 0.05, as shown in Table 1. The top five up- and downregulated DEGs with the highest networking scores are shown in Table 2, and their respective KEGG pathways are presented in Table 3. Even though the top five DEGs presented the highest networking scores, DCAF4, QKI, and TGOLN2 showed no KEGG involvement. On the other hand, NDUAF2, SDHD, and PTGS2 were involved in many pathways. Interestingly, COX10, one of the highest networking proteins, held the number one spot among all the upregulated DEGs in the resistant patient group compared to the sensitive group. Finally, we investigated the influences of the top five DEGs on CRC patient survival using the Human Protein Atlas database (https://www.proteinatlas.org/ (accessed on 18 May 2022)) [32]. We compared the 5-year survival times of colon adenocarcinoma patients and rectum adenocarcinoma patients, as displayed in Table 4. We noticed that the 5-year survival time substantially depended on cancer location. Importantly, LSM5 and DACF4 upregulation in resistant patient samples seem to be unfavorable factors in colon adenocarcinoma patients, leading to a higher mortality rate. Surprisingly, NDUFA2, SDHD, and COX10, which were upregulated in resistant patients, positively influenced the patient survival rate. On the other hand, in rectum adenocarcinoma, the upregulation of NDUFA2 was a highly unfavorable factor, presenting a 19% 5-year survival rate in comparison to 67% for the NDUFA2 low-expression cohort. Furthermore, in the case of the top five downregulated DEGs in resistant CRC patients, a low expression level of PTGS2 was considered an unfavorable factor for both colon and rectum adenocarcinoma. In contrast, for only rectum adenocarcinoma, QKI and RBM8A were recognized as such.

To better characterize the CRC cell lines and their respective SN-38-resistant variants, we analyzed changes in the mRNA expressions of the top up- and downregulated DEGs and the best networking proteins identified in patients’ samples. The data presented in Figure 4A,B clearly show that the data from the CRC cell line analysis partially overlapped with the data obtained from the CRC patient sample analysis. SN-38-resistance-related changes in the mRNA profiles of LoVo lines presented a significant resemblance to the changes in the mRNA levels of NDUAF2, DCAF4, and PTGS2. Interestingly, NDUAF2 and PTGS2 were two of the three most influential tested proteins involved in several important pathways (Table 3). HCT116 and its SN-38-resistant variant, in comparison to the patient samples, presented similar (statistically significant) changes in the mRNA expressions of multifunctional PTGS2 and LSM5, which are involved in RNA degradation and the spliceosome mechanism. The cell lines also presented a similar tendency to that observed in CRC patient samples (yet not statistically significant change) in the mRNA expression profile of RBM8A, which is also involved in spliceosome and RNA transport. The same expression tendency was observed in the case of NDUFA2, but the mRNA expression profile of SDHD demonstrated an inverse expression profile. In HT29, similar to the patient samples, the mRNA expressions of TIPM1 and TGOLN2 were downregulated in the SN-38-resistant variant. The expression of NDUFA2 was also downregulated in the SN-38-resistant variant; however, its expression was upregulated in resistant patients. Surprisingly, COX10, the number one among upregulated DEGs and one of the highest networking proteins, was not upregulated in any SN-38-resistant variants.

Next, we performed a clustering analysis using the mRNA expression profiles of the top DEGs, as shown in Figure 5. The results indicated that HT29 and HTC116, with their respective SN-38-resistant variants, differed enough to establish their phylogenetic groups but were closely located in two respective arms. Interestingly, LoVo and LoVo SN-38 cell lines presented the highest diversity inside the resistant–parental groups and among all the tested cell lines, located separately on different arms of a phylogenetic tree. The expression patterns of DCAF4, COX10, and TGOLN2 (Figure 4 and Figure 5) presented minor differences inside each cell line pair, i.e., parental and irinotecan-resistant variants; however, they substantially differed between cell line sets. This observation could be utilized as an identification tool to separate CRC cell lines. Furthermore, it could provide a deep molecular differentiation pattern for patient-derived samples and potentially generate a new CRC classification type [33].

To analyze the mutual changes between different DEGs, we correlated the mRNA expression profiles of DEGs and the highest networking proteins. A Pearson correlation matrix was performed, as shown in Figure 6. This analysis showed that, in LoVo (pooled parental and the respective SN-38-resistant variant), similar to the patient samples, most of the top five upregulated DEGs (NDUFA2, SDHD, LSM5, and DCAF4) presented positive correlations (yet not statistically significant in all cases). COX10 was positively correlated only with NDUFA2 and DCAF4. Similar to the patient samples, a negative correlation was observed among RBM8A, TGOLN2, and PTGS2. In the cases of HT29 and HTC116, the cell line data were ambiguous, as NDUFA2, SDHD, LSM5, DCAF4, COX10, RBM8A, TIMP1, QKI, TGOLN2, and PTGS2 presented mixed correlations with one another.

EMT was proved to play a role in acquiring chemoresistance to several anticancer drugs [34]. Observation of the LoVo cell-line shift toward the mesenchymal phenotype upon the acquisition of SN-38 resistance encouraged us to analyze EMT phenotypes in FOLFIRI-sensitive and -resistant patient cohorts (GSE62080, *n* = 21). First, using the mRNA levels of four primary EMT markers (E-cadherin (CDH1), N-cadherin (CDH2), vimentin (VIM), and fibronectin (FN)), we divided the patient samples into three groups: strongly epithelial (E), partially mesenchymal (E/M), and mesenchymal (M). However, all the subgroups presented elevated expression levels of CHD1, suggesting an epithelial, well-differentiated phenotype. This observation can be explained by the fact that the GSE62080 dataset comprised cancer samples diagnosed at early stages with no metastatic occurrences. Thus, we assigned the patient samples to the E or E/M groups. (Figure 7A). Interestingly, the sample distribution proved that FLOFIRI-resistant E/M samples presented the upregulation of FN and CDH2 rather than VIM (Figure 7A). Nevertheless, in the resistant cohort (*n* = 12), seven samples were qualified in the E/M group and five in the E group. In contrast, in the sensitive cohort (*n* = 9), three samples were attributed to the E/M group and six to the E group (Figure 7B). These results showed a tendency of the resistant cohort to acquire a more mesenchymal phenotype. Furthermore, this heterogeneous composition of the E and E/M groups, consisting of both resistant and sensitive patient samples in each phenotypical group, partially explained why no statistically significant upregulation of the EMT markers between the resistant and sensitive cohorts was observed in the GSE62080 dataset.

Since the GSE62080 dataset included a low number of patient samples, we decided to apply the GSE18105 dataset to analyze the mRNA expression levels of the previously selected top networking DEGs (upregulated: NDUFA2, SDHD, LSM5, DCAF4, and COX10; downregulated: RBM8A, TIMP1, QKI, TGOLN2, and PTGS2) in CRC patient samples. The GSE18105 (https://www.ncbi.nlm.nih.gov/geo/geo2r/?acc=GSE18105 (accessed on 18 May 2022)) database was composed of mRNA profiles (*n* = 111) obtained using oligonucleotide microarrays of normal tissue and CRC patient primary tumors in different stages. Similar to the previous analysis, the samples were divided into E, E/M, and M groups using the mRNA levels of four primary EMT markers: E-cadherin, N-cadherin, vimentin, and fibronectin. Only six out of the ten tested proteins, i.e., NDUFA2, SDHD, LSM5, COX10, QKI, and TGOLN2, presented statistically significant changes in the mRNA expressions among the E, E/M, and M patient groups (Figure 8A). We observed that in the M and E/M groups, similar to the resistant patient group, NDUFA2, SDHD, LSM5, and COX10 were upregulated, whereas TGOLN2 was downregulated; additionally, the mRNA levels of QKI were upregulated in the E/M and M phenotype groups, unlike that observed in the resistant patient group. The upregulation of NDUFA2, SDHD, and COX10 in the M and E/M groups was interesting since those DEGs are involved in oxidative phosphorylation (OXPHOS). Several experiments proved that human CRC exhibited higher rates of oxidative phosphorylation than large intestine normal cells [35]. Finally, in the current study, we confirmed our previous observation that ABCC4 was upregulated under EMT [18], as well as in cells presenting the mesenchymal phenotype (M and E/M groups). In contrast, the expressions of ABCG2 were slightly (yet statistically significant) downregulated in the M and E/M groups (Figure 8B). We also showed that ABCC1 correlated with the mesenchymal phenotype, but ABCC5, upregulated in HT29 SN-38-resistant variants, was downregulated in both the E/M and M patient groups. Finally, to prove that our E, E/M, and M groups were composed correctly, we also tested the mRNA levels of LGR5 [31], which were significantly upregulated in both the E/M and M groups compared to E.

## 4. Discussion

Approximately 50–60% of patients diagnosed with CRC eventually develop metastatic disease. Most often, metastases develop after first-line therapy treatments for local disease; thus, drug resistance is still the central problem in CRC treatment [36,37]. Gene profiling, among other methods, has been used to identify genes involved in the progression and the prognosis of CRC or to establish a current CRC classification based on four molecular subtypes (CMS1-CMS4) [38,39]. Previously, we identified NMU as a predictive marker of CRC invasiveness [40]. In the current study, to bridge the gap between in vitro and in vivo studies, we combined and compared the expression profiles of cell lines and patient samples from a publicly available database to select new candidate genes for irinotecan resistance. Many attempts have been made to identify predictive chemotherapy biomarkers of treatment response and resistance in CRC. However, most of these biomarkers, except for the KRAS and BRAF genes, are not accurately predicting treatment response. As such, additional studies are urgently needed to identify and validate novel biomarkers to improve therapy for CRC patients [39].

CRC cell lines have previously been shown to recapitulate primary tumors’ mutational and transcriptional heterogeneity [41,42]. Various CRC cell lines resistant to SN-38 have also been generated to understand the irinotecan resistance mechanism. Drug-resistance-associated molecules have been identified by analyzing parental irinotecan-sensitive cells and irinotecan-resistant cells with molecular biology and cellular methods [43]. Our study examined the expression profiles of three cell lines, HT29, HCT116, and LoVo, and their SN-38-resistant variants. We used these three cell lines since they present various genetic, epigenetic, and molecular subtypes, and to some extent, they may reflect the molecular heterogeneity of CRC [41]. Since it has been demonstrated that the assessment of multiple biomarkers provides a more accurate prediction of drug response than a single biomarker, in our study, we analyzed the main enzymes involved in irinotecan transport and processing and, additionally, the main EMT players, as the mesenchymal phenotype is associated with irinotecan resistance [18,39]. First, we concluded that SN-38 treatment changed the expression profiles of all the tested cell lines, irrespective of their molecular backgrounds and origins. The expressions of primary SN38 and irinotecan transporters, i.e., ABCB1, ABCC1, and ABCC2, were significantly upregulated in most SN-38-resistant variants, and the expression profiles of ABC transporters were specific to each cell line. This observation aligns with many other reports on in vitro studies concerning ABC transporter function in cancer [21]. We also observed various changes in the expression profiles of the primary EMT markers in each tested cell line. EMT is the best-known cause of tumor cell plasticity, which appears to influence sensitivity to various chemotherapeutic drugs, and EMT is an essential regulator of ABC transporters [44]. According to our previous data, the phenotype conversion induced by the Snail transcription factor decreased ABCG2 and increased the ABCC4 expression level in HT29 cells [40]. In the current study, we observed that the LoVo-SN-38-resistant variant, which exhibited an evident mesenchymal phenotype, showed the upregulation of ABCG2 expression. A possible explanation for this inconsistency could be that LoVo and HT29 present different genetic profiles and belong to various CRC molecular subtypes—LoVo to CMS1 and HT29 to CMS3—or that LoVo-SN-38-resistant variants acquire mesenchymal features through SN-38 treatment. Our analysis of the primary enzyme responsible for irinotecan activation and deactivation indicated the different expression profiles among the tested lines. The changes in their expressions are relevant in the case of irinotecan administration; however, in the case of the analysis of CRC cell lines, their expression profiles are less attractive since the majority of their activity related to irinotecan in the body is located in the liver [8]. In our tested CRC cell lines, they could process the inactive irinotecan metabolite NPC into active SN-38. A further clustering analysis confirmed that all the SN-38-resistant variants presented significantly different expression profiles than their respective parental cell lines.

To verify our above observation, we analyzed the expression profiles of patients with advanced colon cancer who were sensitive or resistant to the first-line treatment of FOLFIRI. The combination of irinotecan with fluorouracil and calcium folinate significantly increased response rates, time to progression, and survival, presenting response rates up to 40–50%, while 5-FU alone showed only a 10–15% response rate. Interestingly, no differences in response rates were noted for tumors treated with 5-FU alone compared to 5-FU plus leucovorin [45,46]. Thus, we could assume that irinotecan is a major therapeutic factor in the FOLFIRI regimen., In the patient samples, no statistically significant changes were observed in the mRNA expression levels of either ABC transporters or irinotecan-processing enzymes between the sensitive and resistant patient groups. We confirmed the upregulation of LGR5, a marker of advanced CRC, in the resistant patient group concomitantly; the expression profiles of EMT players from this group demonstrated the epithelial phenotype. The epithelial phenotype may be related to 5-FU resistance [47], as patients were treated with 5-FU within the FOLFIRI regimen.

We analyzed the top 250 differently expressed genes (DEGs) between the resistant and sensitive patient groups using a protein–protein interaction (PPI) network to identify new candidate genes for irinotecan resistance. Then, we selected the five upregulated (NDUFA2, SDHD, LSM5, DCAF4, and COX10) and downregulated (RBM8A, TIMP1, QKI, TGOLN2, and PTGS2) DEGs with the highest networking scores. These genes have not been reported as irinotecan-resistance-associated to date; thus, we checked their utility as prognostic markers. Using the KEGG database, we showed the involvement of the selected DEGs in significant cellular pathways. Oxidative phosphorylation was the most-upregulated pathway, while the processes related to RNA processing (RNA spliceosome, surveillance, transport, and degradation) were the most-downregulated. These findings are consistent with the previous observation that cells, to survive under chemotherapy treatment, reduce transcription rates and increase energy demands. Several experiments proved that human CRC cells exhibited even higher rates of oxidative phosphorylation than large intestine normal cells [48]. The elevated production of ATP may be connected to the increase in drug efflux operated by ABC transporters or the increased migration of advanced CRC cells. Irinotecan-resistant non-small cell lung cancer (NSCLC) cells are characterized by increased oxidative phosphorylation, and treatment with gossypol (pan-ALDH inhibitor) and phenformin (OXPHOS inhibitor) reversed irinotecan resistance in a tested xenograft model of human NSCLC [35]. A CRC patient 5-year survival analysis showed that the prognostic values of the selected DEGs depended on cancer location, i.e., colon vs. rectum, and they could be favorable or unfavorable factors. These results are difficult to discuss; however, they align with current knowledge that, in the case of CRC, actual tumor location is one of the most important prognostic factors [39].

To integrate the obtained data and to facilitate the selection of cell lines as appropriate research models, we reversed the standard data flow axis from “cell lines to patients” to “patient-derived data to cell lines”. Although in vivo irinotecan resistance mechanisms are more complicated than in vitro mechanisms, investigating such mechanisms in vitro can provide strategies to overcome drug resistance in CRC [21]. The top five networking up- and downregulated DEGs (NDUFA2, SDHD, LSM5, DCAF4, COX10, RBM8A, TIMP1, QKI, TGOLN2, and PTGS2) were differently expressed in the majority of the irinotecan-resistant variants of the CRC cell lines and often presented a reversed tendency from that observed in vivo. However, this gene set could be utilized to analyze cell line heterogeneity and, as such, the utility of a specific in vitro model. Interestingly, the expression patterns of DCAF4, COX10, and TGOLN2 presented the best tool set to identify and separate cell lines with their respective irinotecan-resistant variants from other pairs of parental–resistant CRC cells lines that were delivered from different cancer locations or stages. This tool could enhance CRC taxonomy and support clinical decisions on therapy selection [33,49]. The obtained data confirmed that in vivo irinotecan resistance mechanisms were more complicated than in vitro mechanisms and suggested that, not one but many, different mechanisms, signaling pathways, and other factors such as other drugs or tumor microenvironment may act simultaneously or complementarily in the development of irinotecan resistance in CRC. We can assume that our results may need to be verified by a specific irinotecan treatment patient cohort. However, in the case of CRC, irinotecan is commonly applied to patients in the FOLFIRI scheme. Our current analysis may also imply why plenty of promising in vitro studies have failed in clinical implementation and why the mechanisms leading to irinotecan resistance are still poorly characterized. However, cell lines still represent a mainstay in functionalizing molecular data, as they allow experimental manipulation, global and detailed mechanistic studies, and high-throughput applications.

Our study showed that by integrating expression profile data, each CRC cell line can be a resource to select relevant models for studies of irinotecan resistance mechanisms.

## 5. Conclusions

After more than two decades of clinical usage, irinotecan represents the backbone of CRC chemotherapy. Although irinotecan resistance mechanisms in vivo are more complicated than those in vitro, investigating the latter is less complex and more cost-effective. Thus, it can provide an easier way to elaborate strategies to overcome overall drug resistance in CRC. Our analysis provided several potential irinotecan resistance markers previously not described: NDUFA2, SDHD, LSM5, DCAF4, COX10, RBM8A, TIMP1, QKI, TGOLN2, and PTGS2. The selected genes and their corresponding mRNA and protein levels could be used clinically to determine the possible outcome of chemotherapy or could be utilized as potential therapeutical targets to overcome irinotecan resistance. We also confirmed that data from publicly available databases, such as GEO or the Human Protein Atlas, are powerful tools to enhance and support in vitro studies. Thus, in a cell line-based study, more than one cell line should be employed, and the obtained data should be compared with patient-derived data.

## Figures and Tables

**Figure 1 biomedicines-10-01720-f001:**
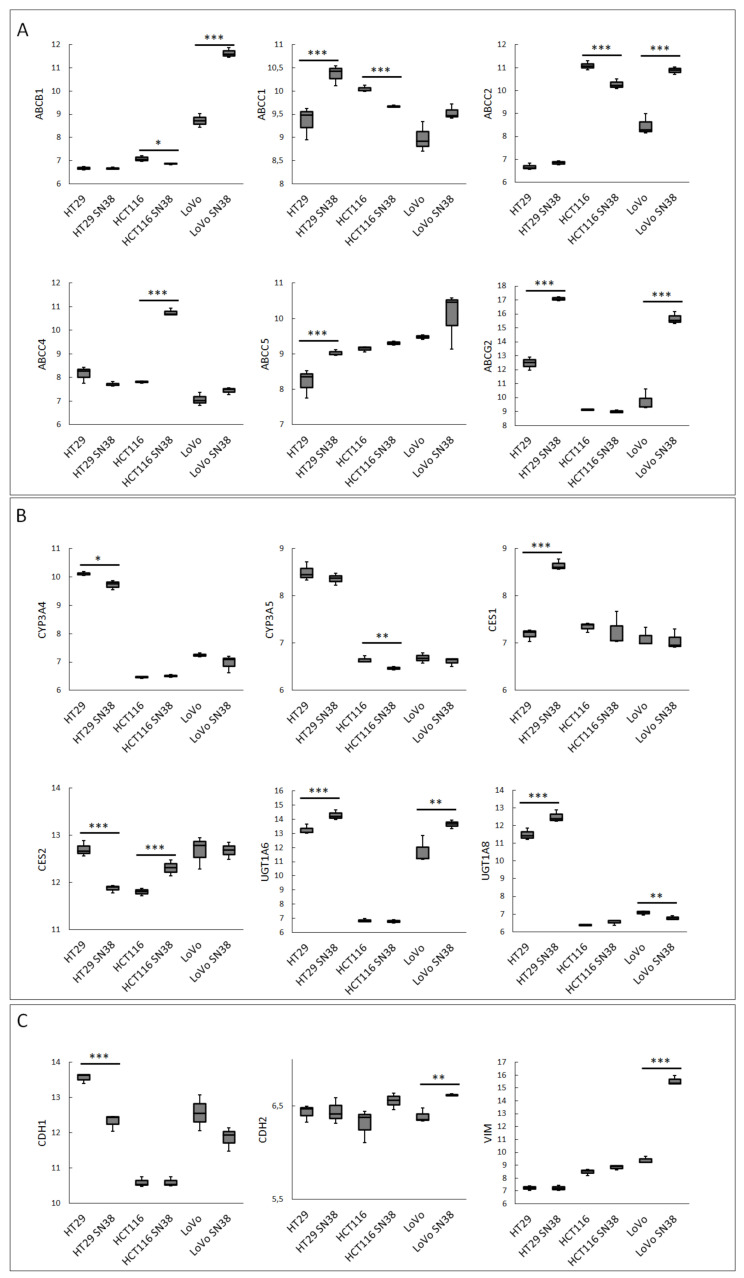
The expression levels of irinotecan-resistance-related proteins in CRC cell lines. Data were obtained from the GEO database of GSE42387 (using GPL16297 Agilent-014850 Whole Human Genome Microarray 4x44K G4112F platform). Changes in mRNA expressions (Y-axis) of (**A**) ABC proteins, (**B**) irinotecan-metabolism-related enzymes, and (**C**) EMT markers were measured in every cell line variant (HT29, LoVo, and HTC116 and their irinotecan (SN-38)-resistant variants HT29 SN-38, LoVo SN-38, and HTC116 SN-38) in triplicate. Normality test (Shapiro–Wilk’s) followed by Mann–Whitney U test (for not normally distributed data) or T-test (for normally distributed data); * *p* < 0.05, ** *p* < 0.005, and *** *p* < 0.001.

**Figure 2 biomedicines-10-01720-f002:**
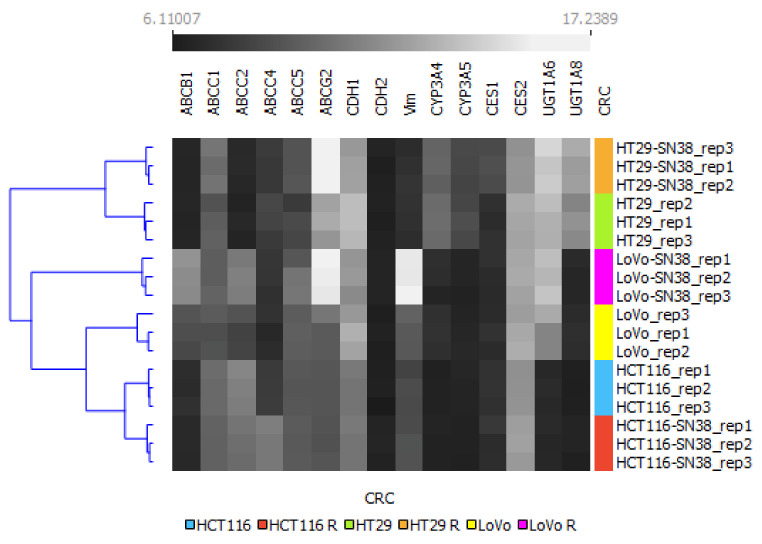
Hierarchical clustering analysis of CRC cell lines using mRNA expression profiles of irinotecan-resistance-related proteins. The expression values between SN-38-resistant variants and corresponding parental cell lines are grayscale; black represents low and white represents high expression values (6.1–17.23). A bidirectional hierarchical clustering heat map was visualized using the Orange data visualization platform.

**Figure 3 biomedicines-10-01720-f003:**
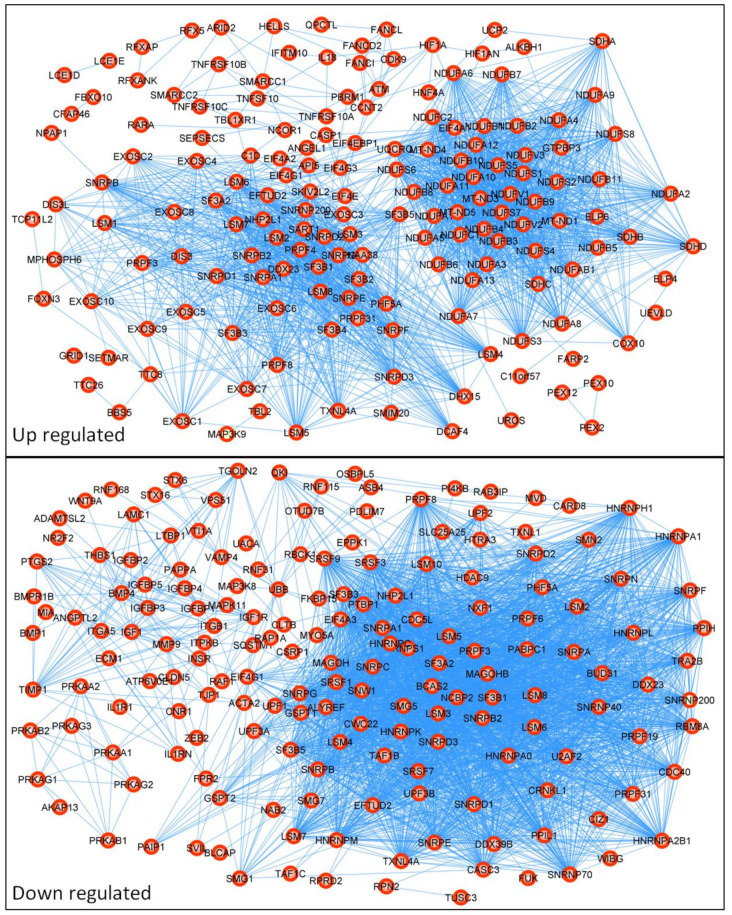
Protein–protein interaction (PPI) network of differentially expressed genes (DEGs). The PPI network was created from 250 differently expressed genes obtained from the GSE62080 database using STRING version 11.0 online software. The PPI pairs were imported into Cytoscape software described in the Materials and Methods section. The lines represent the interaction relationships between nodes.

**Figure 4 biomedicines-10-01720-f004:**
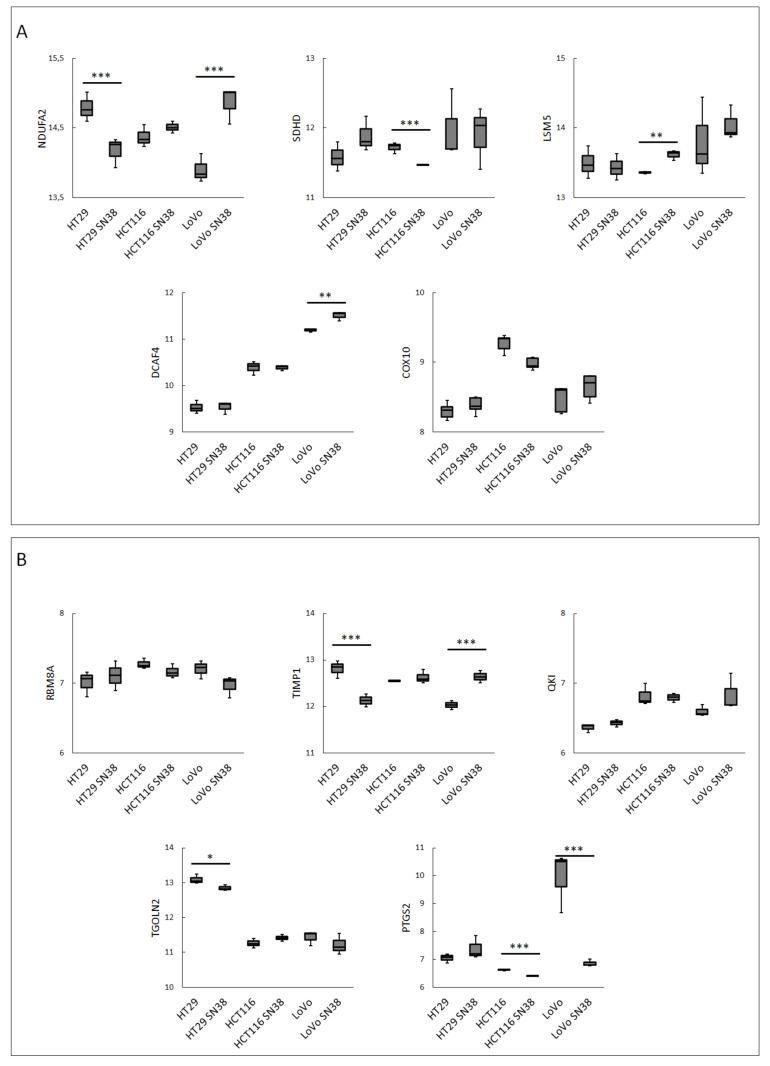
The mRNA expression levels in CRC cell lines of the 5 top networking up- (**A**) and down- (**B**) regulated DEGs from FOLFIRI-resistant patient cohort. Data were obtained using the GOE database: GSE42387 (GPL16297 Agilent-014850 Whole Human Genome Microarray 4x44K G4112F platform). Normality test (Shapiro–Wilk’s) followed by Mann–Whitney U test (for not normally distributed data) or T-test (for normally distributed data); * *p* > 0.05, ** *p*> 0.005, *** and *p* > 0.001.

**Figure 5 biomedicines-10-01720-f005:**
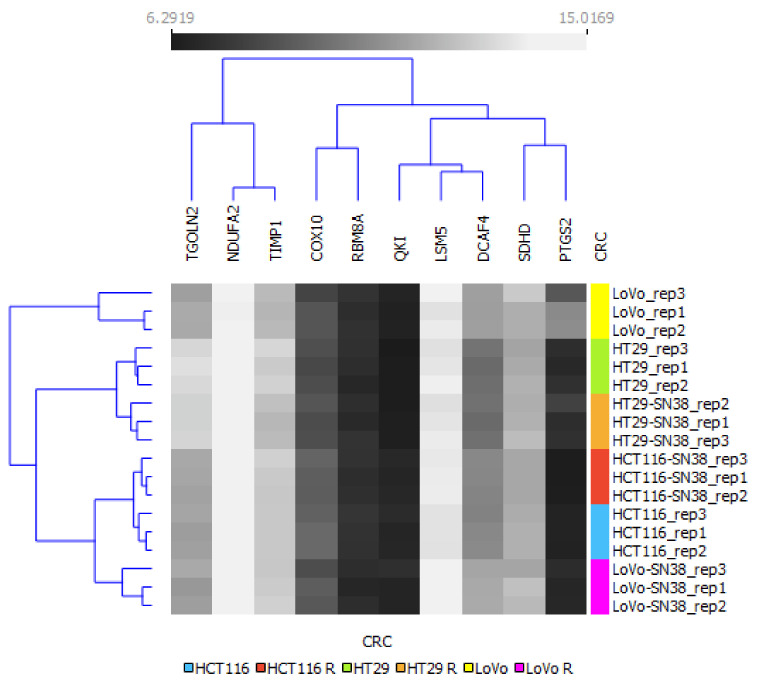
Hierarchical clustering analysis of CRC cell lines using mRNA expressions of top networking DEGs from FOLFIRI-resistant patient cohort. The expression values between corresponding irinotecan-resistant and parental CRC cell lines are presented in grayscale; black represents low and white represents high expression values (6.29–15.02). A bidirectional hierarchical clustering heat map was visualized using the Orange data visualization platform.

**Figure 6 biomedicines-10-01720-f006:**
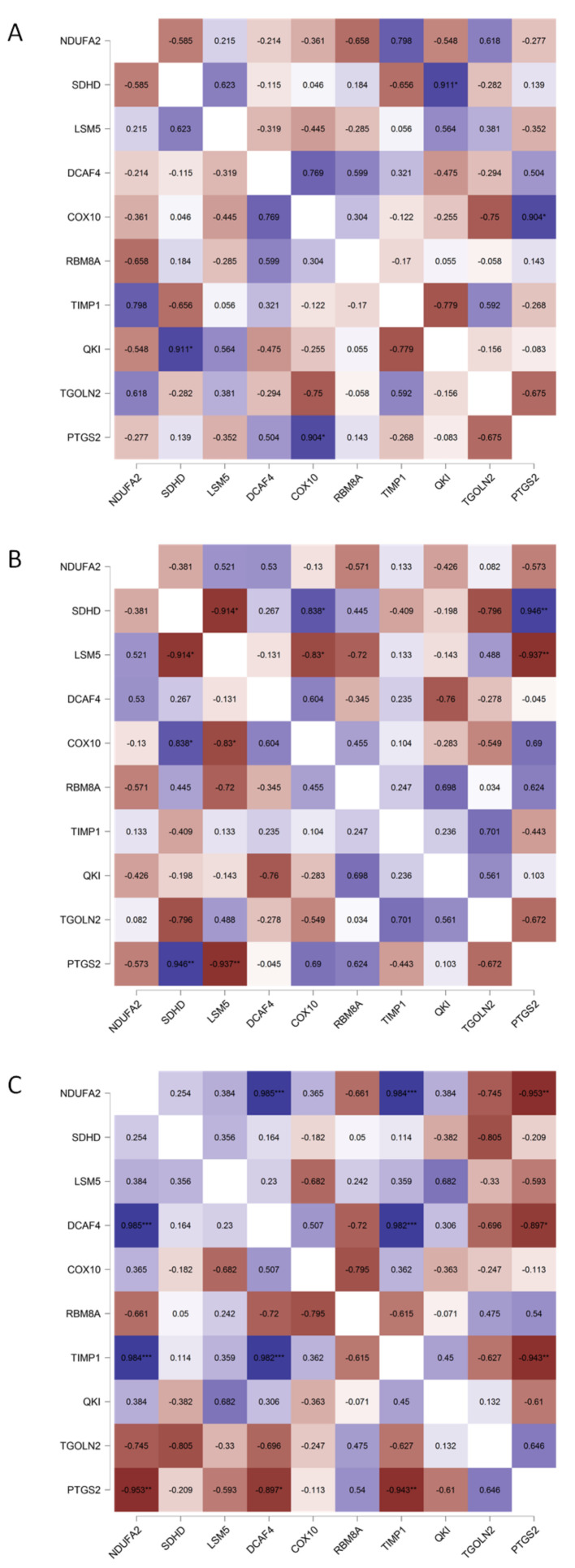
Pearson correlation matrix of FOLFIRI-resistance-related top networking DEGs. mRNA expression data of parental and corresponding resistant variants of CRC cell lines (GSE42387) (**A**) HT-29, (**B**) HTC116, and (**C**) LoVo were pooled to create a Pearson correlation matrix using JASP 0.14.1.0 software; * *p* < 0.05, ** *p* < 0.005, and *** *p* < 0.001.

**Figure 7 biomedicines-10-01720-f007:**
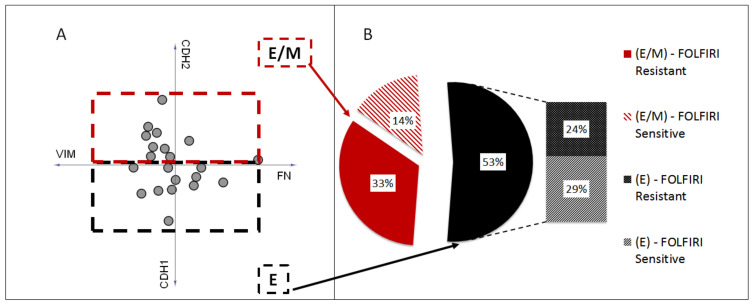
EMT phenotypes of FOLFIRI-resistant and -sensitive patients. Patient samples (*n* = 21) from GSE62080 were attributed to two groups, strongly epithelial (E) and partially mesenchymal (E/M), based on the mRNA levels of four primary EMT markers, E-cadherin (CDH1), N-cadherin (CDH2), vimentin (VIM), and fibronectin (FN), using Orange software (**A**). Sample distribution among E and E/M groups (**B**).

**Figure 8 biomedicines-10-01720-f008:**
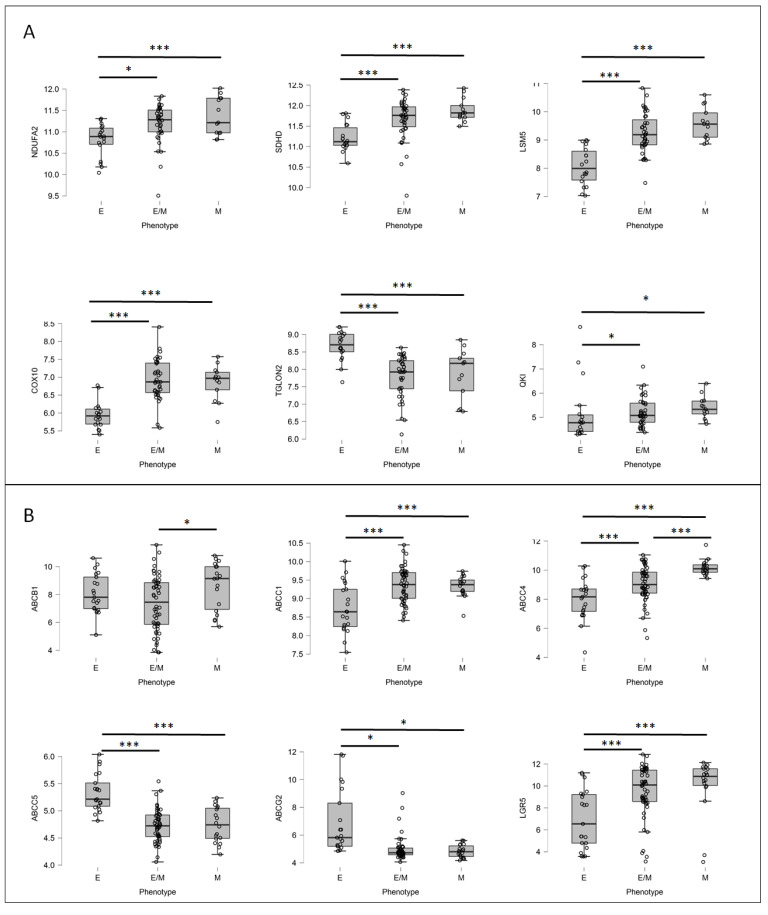
The mRNA expression levels of FOLFIRI-resistant DEGs (**A**) and major ABC proteins (**B**) in various EMT states in patient samples. Patient samples (n = 111) from GSE18105 were divided into three groups, strongly epithelial (E), partially mesenchymal (E/M), and mesenchymal (M), based on mRNA levels of 4 main EMT markers, E-cadherin (CDH1), N-cadherin (CDH2), vimentin, and fibronectin, using Orange software. Next, mRNA expressions of irinotecan-resistant-related DEGs (**A**) and ABC proteins (**B**) were visualized for each patient in their respective groups using JASP 0.14.1.0 software. Normality test (Shapiro–Wilk’s) followed by Mann–Whitney U test (for not normally distributed data) or T-test (for normally distributed data); * *p* < 0.05, *** *p* < 0.001.

**Table 1 biomedicines-10-01720-t001:** KEEG pathway analysis of PPI network. Visualization of top dysregulated pathways by up- and downregulated DEGs in resistant patients samples from GSE62080 database using KEGG PATHWAY database (https://www.kegg.jp/kegg/pathway.html (accessed on 18 May 2022)).

Gene Name	Count in Network	FDR
**Upregulated Genes**		
Oxidative phosphorylation	48 of 131	7.97 × 10^−54^
RNA degradation	23 of 77	3.48 × 10^−24^
Retrograde endocannabinoid signaling	42 of 148	1.54 × 10^−43^
Spliceosome	34 of 130	3.52 × 10^−34^
Thermogenesis	50 of 228	3.39 × 10^−47^
Citrate cycle (TCA cycle)	4 of 30	0.0028
Metabolic pathways	52 of 1250	2.16 × 10^−17^
**Downregulated genes**		
Spliceosome	60 of 130	4.04 × 10^−72^
mRNA surveillance pathway	21 of 89	4.61 × 10^−19^
SNARE interactions in vesicular transport	4 of 33	0.0044
RNA transport	19 of 159	1.75 × 10^−12^
RNA degradation	8 of 77	4.26 × 10^−5^
Adipocytokine signaling pathway	7 of 69	0.00018
FoxO signaling pathway	12 of 130	9.57 × 10^−7^
AMPK signaling pathway	10 of 120	1.89 × 10^−5^
Glucagon signaling pathway	7 of 100	0.0012
Oxytocin signaling pathway	10 of 149	8.76 × 10^−5^
Insulin signaling pathway	9 of 134	0.00021
Apelin signaling pathway	9 of 133	0.00021
Tight junction	11 of 167	4.24 × 10^−5^
Endocrine resistance	5 of 95	0.0221
Signaling pathways regulating pluripotency of stem cells	7 of 138	0.0059
microRNAs in cancer	7 of 149	0.0080
mTOR signaling pathway	7 of 148	0.0080
Proteoglycans in cancer	9 of 195	0.0024
Rap1 signaling pathway	9 of 203	0.0030
Leukocyte transendothelial migration	5 of 112	0.0401
Focal adhesion	8 of 197	0.0085
Thermogenesis	9 of 228	0.0059
Herpes simplex infection	7 of 181	0.0206
PI3K-Akt signaling pathway	10 of 348	0.0231

**Table 2 biomedicines-10-01720-t002:** Top DEGs that had interactions in the protein–protein interaction (PPI) network (GSE62080 database).

Gene Name	Gene	Betweenness Centrality	Degree
NADH Ubiquinone Oxidoreductase Subunit A2	NDUFA2	0.009082	48
Succinate Dehydrogenase Complex Subunit D	SDHD	0.013266	41
LSM5 Homolog, U6 Small Nuclear RNA, and MRNA-Degradation-Associated	LSM5	7.02 × 10^−4^	36
DDB1 and CUL4-Associated Factor 4	DCAF4	0	25
Cytochrome C Oxidase Assembly Factor Heme A: Farnesyltransferase	COX10	4.08 × 10^−5^	24
RNA-Binding Motif Protein 8A	RBM8A	0.007638	75
TIMP Metallopeptidase Inhibitor 1	TIMP1	0.031803	20
KH Domain-Containing RNA-Binding	QKI	0.035524	17
Trans-Golgi Network Protein 2	TGOLN2	0.062555	16
Prostaglandin-Endoperoxide Synthase 2	PTGS2	0.018124	14

**Table 3 biomedicines-10-01720-t003:** Involvement of the top networking DEGs in significant KEGG pathways. Gene enrichment analysis of the DEGs involved in resistance of CRC patients treated with FOLFIRI (GSE62080 database) and their involvements in significant KEGG pathways using KEGG PATHWAY database (https://www.kegg.jp/kegg/pathway.html (accessed on 18 May 2022)).

	NDUFA2	SDHD	LSM5	DCAF4	COX10	RBM8A	TIMP1	QKI	TGOLN2	PTGS2
Arachidonic acid metabolism										x
Biosynthesis of cofactors					x					
Biosynthesis of secondary metabolites		x			x					
Butanoate metabolism		x								
Carbon metabolism		x								
Chemical carcinogenesis										x
Citrate cycle (TCA cycle)		x								
C-type lectin receptor signaling pathway										x
HIF-1 signaling pathway							x			
Human cytomegalovirus infection										x
Human papillomavirus infection										x
IL-17 signaling pathway										x
Metabolic pathways	x	x			x					x
Microbial metabolism in diverse environments		x								
mRNA surveillance pathway						x				
NF-kappa B signaling pathway										x
Oxidative phosphorylation	x	x			x					
Oxytocin signaling pathway										x
Pathways in cancer										x
Regulation of lipolysis in adipocytes										x
Retrograde endocannabinoid signaling	x									x
RNA degradation			x							
RNA transport						x				
Spliceosome			x			x				
Thermogenesis	x	x			x					
TNF signaling pathway										x
VEGF signaling pathway										x

**Table 4 biomedicines-10-01720-t004:** The prognostic relationships between high and low expressions of specific genes involved in drug resistance to the overall survival of colon and rectum adenocarcinoma patients. Five-year survival factor obtained from the Human Protein Atlas (https://www.proteinatlas.org (accessed on 18 May 2022)) [32].

	Colon Adenocarcinoma		Rectum Adenocarcinoma	
	5-Year Survival			
Gene	High	Low	High	Low
NDUFA2	72%	58%	19%	67%
SDHD	71%	52%	65%	19%
LSM5	43%	69%	50%	41%
DCAF4	61%	71%	61%	45%
COX10	71%	57%	66%	25%
RBM8A	57%	71%	60%	23%
TIMP1	53%	69%	34%	89%
QKI	64%	63%	57%	0%
TGOLN2	64%	63%	62%	47%
PTGS2	69%	59%	59%	20%

## Data Availability

Not applicable.

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
