# Peer review of "Integrated Bioinformatics Analysis of the Hub Genes Involved in Irinotecan Resistance in Colorectal Cancer"

_biomedicines, 2022, doi:10.3390/biomedicines10071720_

Round 1

Reviewer 1 Report

Dear Authors

In the research paper entitled “Integrated bioinformatics analysis of the molecular landscape of irinotecan resistance in vitro and in vivo” it is shown that in irinotecan-resistance CRC cell lines and in the analysis of GEO profile the NDUFA2, SDHD, LSM5, DCAF4, COX10 RBM8A, 16 TIMP1, QKI, TGOLN2 and PTGS2 genes are deregulated. Furthermore, these results indicated that in vitro models are limited respect to patient-derived models. Therefore the in vitro model can only be used as a molecular model of irinotecan resistance, but it is difficult to translate these results as resistance models in CRC patients.

In my opinion that  there are some questions you could answer:

1) For a better understanding of the results related to the analysis of GEO profiles the authors should enter a descriptive table of the patient clinical-pathological characteristics.

2) The authors should demonstrate that in irinotecan-resistant CRC cell lines the modulation of the detected genes (siRNAs) restores sensitivity to treatment again.

3) Regarding to material and methods the authors must include a separate paragraph for cells cultures and transcriptomic assay.

4) The manuscript needs a revision of the grammar.

Author Response

Response to Reviewer #1 Comments:

Comment #1: For a better understanding of the results related to the analysis of GEO profiles the authors should enter a descriptive table of the patient's clinical-pathological characteristics.

Response #1: Patient clinical characteristics were included in the original paper cited as ref.27. We improved the description of GEO profiles by adding the URL address in lines 126-131 in the Microarray data processing and analysis section.

Comment #2:  The authors should demonstrate that in irinotecan-resistant CRC cell lines the modulation of the detected genes (siRNAs) restores sensitivity to treatment again.

Response #2: We thank the Reviewer for this valuable comment. We agreed that applying siRNA to confirm the role of selected genes in irinotecan resistance would increase the significance of obtained results markedly. However, the main aim of our study was to show how to use available public data and bioinformatics tools to find the genes for further detailed analysis. We plan to investigate the role of selected genes in irinotecan resistance in the future. Ten days revision period is a short time to conduct experiments with siRNA and further analyze and validate the effects of gene expression silencing.

Comment #3: Regarding to material and methods the authors must include a separate paragraph for cells cultures and transcriptomic assay.

Response #3: We modified the description Microarray data processing and analysis section to clarify the data origin. We also added the additional information derived from the original paper and URL link for better access to analyzed data, lines 118-124.

Comment #4: The manuscript needs a revision of the grammar.

Response #4: English grammar was revised

Reviewer 2 Report

Specific comments to the authors

The authors Jakub Kryczka and Joanna Boncela of the submitted manuscript „Integrated bioinformatics analysis of the molecular landscape of irinotecan resistance in vitro and in vivo.” studied the molecular status as biomarker of irinotecan resistance. They applied a sophisticated bioinformatic in-silico approach of mRNA expression profiles of CRC cell lines (HT29, HTC116, LoVo, and their respective irinotecan-resistant variants) in comparison to patient samples.

In summary, based on their investigational approach the authors could select and validate the new candidate 11 genes for validation of irinotecan resistance: NDUFA2, SDHD, LSM5, DCAF4, COX10 RBM8A, 16 TIMP1, QKI, TGOLN2 and PTGS2. Therefore, the authors postulated (i) in-vitro analysis can be useful to analyze specific mechanisms on their molecular level, (ii) whereby these in-vitro findings does not reflect the complicated drug resistance network observed in patients.

Overall, the manuscript give some interesting information of genes potentially involved in chemoresistance in CRT looking on in-vitro or in-situ molecular data. Although the manuscript (including presentation) is mostly comprehensible and convincing, the switch from in-vitro to in-situ and vice versa is largely confusing. The methods are mostly well described. Although the results and discussion are mostly clear presented, the major concerns relate to (i) the missing thread throughout the manuscript and to (ii) question of comparability of the molecular data of the CRC cell lines and patients. Therefore, the authors (see specific comments) must perform some major changes to improve the manuscript. In conclusion, the presented data are interesting. After incorporating the mentioned specific comments (see below) the manuscript has the potency to be accepted.

Specific comments

Title: The title is rather unspecific and does not reflect the content of the submitted manuscript (especially the relation to colorectal cancer). Please change adequately.

Abstract: The conclusion is largely unspecific and represents more a common statement of use of an integrated bioinformatics analysis.

Introduction: The sentence “The mechanism of irinotecan resistance is still inconclusive and requires further investigation.” is contradictory regarding the next two paragraphs.

Material&Methods: The mentioned link of “All data were processed using the GEO2R 124 online analytical tool, which uses the R language (https://doi.org/10.1371/jour-125 nal.pone.0180616 ) is not correct (see https://www.ncbi.nlm.nih.gov/geo/geo2r/ )

Results:

# Figure 1: It is not clear why the authors does focus on ABC proteins, metabolism-related enzymes and EMT as well as these genes. Please indicate the “irinotecan”-resistance cell-lines in the figures. What is the difference between the findings in figure 1 and the publication under ID#18. Please clarify.

# Figure 2: It is surprising that the “irinotecan”-resistance cell-lines are not grouped on the same side? What variable(s) or algorithms are used for the hierarchical clustering analysis.

# Supplementary Figure S1: The authors should compare the same genes of figure 1 to compare the in-vitro and the in-situ “situation”.

# Figure 3, Table 1 to 3: It is not clear which database is used for this bioinformatic analysis. Please clarify. What is the definitive relation of table 1 und 2 as well 3. The common thread is unfortunately not apparent. 

# Table 4: Instead of low and high 5-year survival probabilities, the authors should show classical survival curves.  

# Figure 4a/b: Why does the authors switch to in-situ data. 

# Figure 5 and 6: What is the additional information of figure 6 to 5? Regarding the genes, a dichotomous of the genes with binary logistic regression could be helpful.

# Figure 7 and 8: It is not clear, why the some of the FOLFIRI resistant DEGs and major ABC proteins are related/correlated to the EMT-phenotype.

Discussion: Overall, the discussion is largely descriptive and narrative. Therefore, the authors should focus on the “own” and relevant data. Furthermore, the authors should discuss the limitations of the presented bioinformatics study in detail (no in-vitro, no in-vivo and no in-situ analysis). Furthermore, the authors analyze SN-38 resistant CRC cell line and compared the findings to two published patient groups with partial low case number treated with FOLFIRI, whereby FOLFIRI is a combination of folinic acid, fluorouracil (5-FU) and irinotecan (Camptosar). Therefore, it is not clear which compound has which effect on chemoresistance. This should/must be discussed, too. How could the interesting findings transferred from a theoretical to a practical view (like clinical setting (drug-development, drug-combination))? Please discuss in short.

Author Response

Response to Reviewer #2 Comments:

Specific comments

Comment #1 : Title: The title is rather unspecific and does not reflect the content of the submitted manuscript (especially the relation to colorectal cancer). Please change adequately.

Response #1: We agree, and we presented more specifically the subject of our study in the title, which is currently as follows:,, Integrated bioinformatics analysis of the hub genes involved in irinotecan resistance in colorectal cancer.”

Comment #2: Abstract: The conclusion is largely unspecific and represents more a common statement of use of an integrated bioinformatics analysis.

Response #2: We thank the Reviewer for this comment. The informatics analysis was the tool used to verify our central hypothesis, and we described them less specifically. We agreed that the abstract could be more specific or detailed, but this section is limited to 200 words. In such limited space, we tried to underline the most important results, which in that case are related to irinotecan resistance in CRC.

Comment #3: Introduction: The sentence “The mechanism of irinotecan resistance is still inconclusive and requires further investigation.” is contradictory regarding the next two paragraphs.

Response #3: To clarify this paragraph, we removed lines 52-53. The sentence mentioned above is the introductory sentence to the paragraph that presents the main body’s mechanism involved in irinotecan processing. Those processes, i.e. metabolism and cellular efflux, are still under investigation regarding their relation to irinotecan resistance, and we also analyzed them in this aspect in our current study. The following two paragraphs in 59-89 describe current knowledge on enzymes and protein involved in the irinotecan body's processing; however, the body's processing and irinotecan resistance are not equivalent to each other, and we concluded that fact at the end of the paragraph:,, Although enzymes and transporters involved in the irinotecan body’s metabolism and disposition are known, their contributions to irinotecan resistance are still poorly understood’’.

Comment #4; Material&Methods: The mentioned link of “All data were processed using the GEO2R 124 online analytical tool, which uses the R language (https://doi.org/10.1371/jour-125 nal.pone.0180616 ) is not correct (see https://www.ncbi.nlm.nih.gov/geo/geo2r/ )

Response #4: We replaced the link with the correct one : (https://www.ncbi.nlm.nih.gov/geo/geo2r/ ) and added ref.28.

Comment #5: Results: # Figure 1: It is not clear why the authors does focus on ABC proteins, metabolism-related enzymes and EMT as well as these genes. Please indicate the “irinotecan”-resistance cell-lines in the figures. What is the difference between the findings in figure 1 and the publication under ID#18. Please clarify.

Response #5: In the introduction section, we describe the hypothetical role of the irinotecan body's processing in irinotecan resistance. In the results section in lines 166-175, we explained why we analyzed the ABC transporters, EMT markers and the enzymes involved in irinotecan processing: In our analysis first, we focused on ABC proteins, as their expression level is considered to have a high impact on overall SN-38 intracellular concentration and is mainly analyzed in CRC cells lines to establish the mechanism of acquired irinotecan resistance [27,29] Figure 1A. Then, we analyzed the expression level of enzymes involved in irinotecan/SN-38 metabolism Figure 1B since our previous studies showed that the expression of ABC transporters in CRC was correlated to cell phenotype and shifts during ongoing EMT [18], and considering that many other reports showed that irinotecan resistance is often related to advanced EMT, we additionally compared the expression level of the significant EMT markers, such as E-cadherin (CDH1), N-cadherin (CDH2) and vimentin (VIM) [30] Figure 1C.

In lines 92-94, the introduction section, we described the standard procedure for generating the irinotecan resistant variants of cell lines: ,, Drug-resistant variants of specific cell lines are obtained by continuous exposure to increasing concentrations of irinotecan itself [21] or its active metabolite SN-38 [20] in the growth media for several months’’. We used the term,, cell lines/SN38’’ as analyzed cell lines were obtained by exposing,,in vitro to gradually increasing SN-38 concentrations for about nine months, generating sub-cell lines with acquired resistance'', lines 119-121. The active metabolite of irinotecan, SN38, is commonly used to prepare the resistance variants of specific cell lines. As we stated in lines 59-89:,, Irinotecan is a prodrug that requires bioactivation to form the active metabolite SN-38’’. This bioactivation is mainly localized in the liver and plasma; thus, in the resistance study based on cell line models, SN28 are commonly used as a resistance generating factor. However, the obtained resistant variants reflect the irinotecan resistance mechanism. We clarified the description in Legend to Fig.1 and in the text using the term ,, irinotecan (SN38) resistant variants.''

In paper ID#18, we used HT29 and HT29 with Snail overexpression (EMT inducing factor) as an experimental model. We analyzed their protein expression changes (ABC protein, among others) and compared their migratory and invasive potential.

Comment #6: # Figure 2: It is surprising that the “irinotecan”-resistance cell-lines are not grouped on the same side? What variable(s) or algorithms are used for the hierarchical clustering analysis.

Response #6: The information concerning the used variables and algorithm were added to lines 203-205.

In our study, we analyzed three CRC cell lines. Many studies showed that the CRC cell lines present high genetic heterogeneity and different genetic background. This fact results from various molecular mechanisms responsible for CRC initiation and progression, e.g. chromosomal instability, microsatellite instability, mutational status KRAS wild or mutated etc. Many studies have been performed to generate the CRC cell lines classification. A recent study based on transcriptional analysis showed that the CRC lines could be divided into four CMS (consensus molecular subtype) subtypes (Consensus molecular subtypes of colorectal cancer are recapitulated in vitro and in vivo models. DOI: 10.1038/s41418-017-0011-5). In our study, each cell line represents a different CMS subtype, genetic background, and phenotype and is derived from various locations in the patient. They reflect typical heterogeneity observed for CRC. We discussed this issue in the discussion section, lines 413-416, 430-432,449-452.

Comment #7: Supplementary Figure S1: The authors should compare the same genes of figure 1 to compare the in-vitro and the in-situ “situation”.

Response #7: Figure S1 was changed according to the Reviewer's suggestion.

Comment #8:Figure 3, Table 1 to 3: It is not clear which database is used for this bioinformatic analysis. Please clarify. What is the definitive relation of table 1 und 2 as well 3. The common thread is unfortunately not apparent. 

Response #8: The database descriptions were added to the Legends to table 1,2 and  3.

Tables 1,2, and 3 show the interplay between signalling pathways and top (five) up-and-down-regulated DEGs. Table 1 presents the changes in the signalling pathway, table 2 – shows the top (five) up-and downregulated DEGs and Table 3 shows the relation of top DEGs to signalling pathways selected in Table 1. We clarified this relation in the manuscript in lines 242-245.

Comment #9: Table 4: Instead of low and high 5-year survival probabilities, the authors should show classical survival curves.  

Response #9: We put the data in the table to present the 5-year survival probabilities in a more readable and cohesive way for readers. Generating so many curves for samples from two locations (colon and rectum), one curve for each gene, makes the data less clear and difficult to compare.

Comment #10: Figure 4a/b: Why does the authors switch to in-situ data. 

# Figure 5 and 6: What is the additional information of figure 6 to 5? Regarding the genes, a dichotomous of the genes with binary logistic regression could be helpful.

We apply the data from patient samples analysis to better characterize CRC cell lines and their respective SN-38 resistant variants. The main goal of our study was to show that the open-source database and informatics tools may improve the quality and reliability of in vitro studies. Our results also demonstrated that database and informatics might help plan, more specifically, the experimental part of in vitro study.

We performed the Pearson correlation matrix to show mutual changes between various DEGs. Figure 6 could be informative for further study to verify selected genes that contribute to irinotecan resistance and their mutual relation.

Comment #11: Figure 7 and 8: It is not clear, why the some of the FOLFIRI resistant DEGs and major ABC proteins are related/correlated to the EMT-phenotype.

Response #10: In the current study, we focused on EMT players and - related phenotypes since, based on our previous and other groups' studies, EMT plays a prominent role in acquiring chemoresistance to several anticancer drugs irinotecan resistance is often related to advanced EMT. EMT is also an essential regulator of ABC transporters. It was demonstrated that the promoters of ABC transporters carry several binding sites for EMT-inducing transcription factors. We mentioned those facts several times in the manuscript, particularly in results ( e.g.line166-170, 348-349) and discussion (e.g.line 441-444) section.

Comment #12: Discussion: Overall, the discussion is largely descriptive and narrative. Therefore, the authors should focus on the “own” and relevant data. Furthermore, the authors should discuss the limitations of the presented bioinformatics study in detail (no in-vitro, no in-vivo and no in-situ analysis). Furthermore, the authors analyze the SN-38 resistant CRC cell line and compared the findings to two published patient groups with partial low case numbers treated with FOLFIRI, whereby FOLFIRI is a combination of folinic acid, fluorouracil (5-FU) and irinotecan (Camptosar). Therefore, it is not clear which compound has which effect on chemoresistance. This should/must be discussed, too. How could the interesting findings transferred from a theoretical to a practical view (like clinical setting (drug-development, drug-combination))? Please discuss in short.

Response #11: We thank the Reviewer. We agreed that the mechanism of irinotecan resistance is complicated to analyze due to its complicated body activation and processing and the fact that irinotecan is commonly applied in combination with other drugs. We briefly discussed FOLFIRI components and their response rate in advanced CRC in lines 463-468, adding ref.45,46 and in lines 515-537. We proposed that cell lines are still good and not expensive experimental models to study a single molecular mechanism. We discussed this in lines 518-537. We modified the discussion section to underline the aspects mentioned above and removed the descriptive part of the discussion section.

Round 2

Reviewer 1 Report

N/A